# How *Jaminan Kesehatan Nasional* (JKN) coverage influences out-of-pocket (OOP) payments by vulnerable populations in Indonesia

Nirwan Maulana[1]*, Prastuti Soewondo[1], Nadhila Adani[1], Paulina Limasalle[2], Anooj Pattnaik[2]

1 ThinkWell Institute, Jakarta, Indonesia, 2 ThinkWell Institute, Washington, D.C., United States of America

* nmaulana@thinkwell.global

**Data Availability Statement:** The data (Susenas) that support the findings of this study are available from Indonesian Central Bureau of Statistics, but

## Abstract

While Indonesia introduced a national health insurance scheme (JKN) in 2014 and coverage has grown to over 80% of the population, Indonesians still spend significant sums out-of-pocket (OOP) for their healthcare–over 30% of current health expenditure (CHE). This study aims to better understand how JKN is influencing OOP payments, especially among the poor and rural, at the range of health facilities. This study uses data from the National Socio-Economic Survey (SUSENAS) in 2018 and 2019, as these surveys started including a question on how much OOP spending a household incurs on health. The results show that households with JKN membership are far less likely than the uninsured to pay OOP for healthcare, and that if they do incur a cost, the magnitude of this cost is much lower among JKN households than uninsured ones. The results also show that JKN households in the two poorest quintiles have a higher probability to not incur *any* OOP (37% and 35%, respectively) compared to those in the wealthier quintiles 4 (32%) and 5 (30%). Poorer JKN households living in the eastern part of Indonesia–the less urbanized and developed regions– experienced the most cost-savings, though largely due to supply-side constraints. In fact, JKN members save more at public primary health care facilities vs. private ones (who often do not contract with JKN) and also save significantly more (over 50%) than uninsured households at both public and private hospitals. The study demonstrates the positive influence JKN has on OOP payments, especially among the poor and rural, but also highlights how the scheme needs to better engage with the growing private sector and invest in infrastructure in rural areas to help secure financial protection for its entire population.

## Introduction

One of the main tenets of achieving universal health coverage (UHC) is to protect individuals from financial burden [1]. This burden is often in the form of individuals being forced to make direct payments out-of-pocket (OOP) for receiving health services. These OOP

restrictions apply to the availability of these data, which were used under license of Thinkwell Institute for the current study, and so are not publicly available. Data is available from: https://silastik.bps.go.id/ by submitting an application via the website. Other researchers will be able to access the data set in the same way as the authors, and the authors do not have special access rights that others do not have.

**Funding:** This work was supported by the Bill & Melinda Gates Foundation under the Strategic Purchasing for Primary Health Care grant (Grant Number: OPP1173304). The funders had no role in study design, data collection and analysis, decision to publish, or preparation of the manuscript.

**Competing interests:** The authors declare no competing interests.

payments are highly regressive and can force the most vulnerable individuals to choose between health care and other necessities as well as often push them below the poverty line [2,3]. To protect individuals from such scenarios, countries have been adopting health financing reforms that pool funds through taxes or pre-paid contributions and purchase services from healthcare providers on behalf of the population, especially the most vulnerable [4,5]. Yet, there is mixed evidence on how these large-scale health insurance schemes in lower-and-middle-income countries (LMICs) have an effect on key equity outcomes, such as OOP payments [6–10].

The Government of Indonesia (GoI) has recently moved in this direction, as it implemented the Jaminan Kesehatan Nasional (JKN) national health insurance program in 2014 by establishing Law No. 40/ 2004 on The National Social Security System. The law mandates all individuals to be covered by JKN irrespective of income or employment status; those working in the formal sector are registered in JKN by their employers, while the poor and near-poor are classified as Subsidized Contribution Recipients (PBI) and have their contributions subsidized by national and district authorities. A challenge in Indonesia is covering the informal sector, who make up a significant proportion of the working age population. Unlike formal sector workers and PBI members, those in the informal sector must enroll in JKN themselves to receive membership cards and pay the monthly contribution fee that corresponds to their group. An explicit objective of JKN is to protect the population from financial shocks, thus enhancing people's wellbeing to lead productive lives and eventually increase national productivity [11,12].

JKN is managed by the Social Security Administering Body for Health (BPJS-K) which is separate from the Ministry of Health (MoH). This agency contracts and pays both public and private providers across the country, though the purchasing policies are set by the MoH. By 2019, JKN covered over 84% of the population with a comprehensive benefits package [13]. Prior to JKN, there were many social insurance schemes managed by multiple institutions, such as PT.ASKES who managed social insurance for civil servants and the poor, PT ASABRI who managed the one for military officers, and also several local government's own schemes. Benefits for these schemes differ from one another, but since JKN implementation, these schemes became integrated, and benefits are standardized. Benefits include curative care for outpatient and inpatient care, prescribed medicines, accommodation for inpatient care, and ambulance service to transfer to hospitals if needed [12]. JKN do not cover services that are considered cosmetics, accidents caused by hobbies, drug or alcohol abuse, results of natural disasters, traditional medicines or treatment, and non-prescribed medicines. In addition, JKN membership can only be used in facilities that are already contracted by BPJS-K. As there are no caps imposed, members are free to access primary health care facilities when sick, however referrals are needed to access secondary and tertiary facilities. The scheme has progressively increased utilization of key services and the proportion of OOP payments to total health expenditure (THE) reduced from 48.5% in 2014 to 32.1% in 2019 [14,15].

There have been several studies that look broadly at the influence of JKN on catastrophic health expenditures and the cost for specific service types (e.g. childbirth, family planning) [16–18]. What is less understood is the direct relationship between JKN coverage and OOP payments, and how this relationship has changed for different population members (e.g. poor versus wealthy), at different types of providers (e.g. public versus private, health center versus hospital), and across different geographic areas (e.g. the more urban western versus the more rural eastern regions). This study aims to address this gap in the literature to better understand from an equity perspective, how JKN coverage impacts OOP payments for the most poor and vulnerable populations across Indonesia.

## Methods

This study uses data from the National Socio-Economic Survey (SUSENAS) in 2018 and 2019. It is nationally representative with household data down to the city and district levels. The survey provides general information about households and household members, such as socio-economic status and demographic indicators (including those related to health), while also covering detailed information about household consumption and expenditure. SUSENAS uses a stratified probability sampling design and is conducted biannually in Indonesia–March and September. The sampling design for March is conducted to fully represent district/ city level, while the September one aims to update socio-economic conditions, so that it uses a different set of samples which could be representative until province level. Both datasets are cross-sectional data and cannot be merged. This study employs the March version as it can be disaggregated up to the city and district levels. After applying sampling weights, the SUSENAS 2019 March version consists of 69,954,912 households and 263,666,217 individuals, while the SUSENAS 2018 March version consists of 71,280,887 households and 266,705,582 individuals. Without the sample weights, the 2018 data consists of 1,131,825 individuals and 295,155 households, while the 2019 data consists of 1,204,466 individuals and 315,672 households. S1 Table captures the descriptive statistics of the data disaggregated by insurance types.

We are using 2018 and 2019 SUSENAS data because these surveys started including a question on how much OOP spending a household incurs on health in a given year. Before 2018, the survey would only measure how much total health expenditure a household incurred. This estimation may include government transfers, OOP payments, and insurance [19]. Hence, previous studies using this data may not accurately capture JKN's impact on OOP payments. By comparing with the 2019 data, we could ensure that data structures and quality are consistent. We found that the two years are consistent and are qualified to conduct a cross-sectional analysis. Furthermore, by using the two years, we could have higher numbers of observations which would be favorable for a more robust result, as it reaches nearer to the normal distribution, and could eliminate heteroskedasticity.

Annual households' OOP expenditure in Susenas includes curative (such as inpatient and outpatient by provider), preventive (family planning, medical check-up, immunization) dan medicines (prescribed, non-prescribed, and traditional medication). Information about non-medical expenditure OOP is only available for transport cost to facilities. However, we exclude this because we only focus on the effect of JKN's benefits on OOP. We also excluded OOP payments for non-prescription medicines, traditional medicines, and other medications, because including these components could distort the estimations since JKN can only be utilized at modern health facilities. The unit of analysis for this study is households accessing healthcare from modern health facilities, therefore we had to drop 103,206 households utilizing traditional healthcare. Taking sampling weights into account, total observations in this study are 141,043,416 households. Stata 16/SE was used to conduct our analyses.

### Measures

**Outcome variables.** The main outcome variable is households' OOP health spending, which SUSENAS defines as healthcare costs paid in cash in the last year. SUSENAS 2018 and 2019 measure this variable on an annual basis. OOP health spending was adjusted in real terms using the ratio of the national average poverty line in a base year (2018) over the district's poverty line for the given year. This adjustment allows comparability across regions [20]. We found unreasonably low annual OOP spending possibly due to data entry errors and respondents' recall bias. We decided that responses below Rp 10,000 ($0.70) are recoded as 0 because this is a typical tariff for outpatient care at public PHC facilities. Around 1% of total

observations had annual OOP health spending below this threshold. We tried different thresholds, but the results did not change significantly.

**Explanatory variables and covariates.** The key explanatory variables are the types of insurance a household owned, categorized as: (1) uninsured households; (2) JKN households; and (3) households with private and mixed insurance ownership. This analysis defines JKN households as a household whose members all hold only JKN. If there is at least one household member who does not own insurance or has mixed insurance (e.g., private and JKN) we categorize them as the third group. By isolating potential effects from other insurance types, we could explore stronger associations between JKN ownership and OOP health spending.

To scrutinize this association more deeply, we interact insurance types with four covariates: (1) wealth quintile; (2) location (urban or rural); (3) medical use (outpatient and inpatient) by provider type (PHC facility or hospital); and (4) year. The first two interactions explore the equity aspect of JKN. The last two interactions could be interpreted as the evolving effectiveness of JKN in reducing OOP payments.

Other covariates included in the analysis are health status and household characteristics. We define health status as at least one household member feeling sick in the past month. This variable can capture households with relatively high OOP health spending due to illness. Ideally, there is a control variable that signals chronic illness, but this is not available in Susenas. We included indicators on the head of a household's demographic characteristics–age, gender, education, and occupation–to control for behavioral heterogeneity across households. Indicators for a household's flooring materials and ownership of a defecation facility were included to capture economic status from an asset-based perspective and household sanitation, respectively. Finally, provincial fixed effects are also included to account for spatial heterogeneity in terms of economic development, health infrastructure, and public health policies. We referenced existing literature with similar research designs to select these covariates, such as Ekman (2007), Zhang et al., (2017), Deb & Norton (2018), and Nugraheni et al., (2020) [16,21–23].

## Statistical analysis

This study uses descriptive statistics and econometrics modelling to understand the association between JKN enrolment on OOP health spending. The main challenge for modeling health expenditures is the violation of the normality condition. Health expenditure data are generally positively skewed and contain a large portion of zero values. In our case, 31% of households have zero OOP health spending. As a result, the analysis cannot use a standard ordinary least squares (OLS) model as it will give inconsistent parameter estimates [24]. It is ideal to use a Two-Part Model (2PM), especially when the choice to consume/spend is not strongly influenced by the amount spent. Instead, the biggest factor on the decision to incur OOP health spending is whether an individual has any health issues or not [25].

2PM consists of two sequential regression models. The first part estimates the likelihood of a household incurring zero or positive OOP payments. The second part estimates the level or intensity of OOP payment, conditional on a household spending anything OOP. S2 Table displays the output for the first and second part. However, Two-Part model's output are difficult to interpret directly because it uses a logarithmic function as the regression model is non-linear. Therefore, we use the *margin* command in Stata to obtain the marginal effect of predictors that is referred to in the Results section. For selected key variables, S3 Table exhibits the marginal results of the regression output. Furthermore, to obtain the unconditional predicted OOP estimate (i.e. how much a household spends OOP when obtaining healthcare), the probabilities of paying OOP from the first part are multiplied by the expected levels of OOP from the second part [26,27]. Our main discussions will be focused on this unconditional estimation

of OOP payments. This is because it is more relevant for policy discussions around the influence of JKN on OOP payments and its implications on access and equity for populations across Indonesia. It is important to assess at the population level, as JKN is a social insurance and covers preventive and promotive services -not only curative- the effect of it may also be seen among households with no illnesses.

The first part is a logit model with the binary dependent variable of zero and positive OOP payments, while the second part is a Generalized Linear Model (GLM) with gamma error distribution and a log link function. Both regressions utilize a robust standard error to control for heteroskedasticity. We run a modified Park test to determine the family distribution, which identifies the relationship between the mean and variance by regressing the log-transformed squared residuals on the log-transformed expected value from a provisional specification, i.e., GLM [28,29]. The test suggests using gamma distribution because the coefficient of the expected value ($\lambda$) was 1.97. On the other hand, most literature suggests using the log link function since covariates for health expenditure data typically act multiplicatively on the expected mean function [22,30–32].

Meanwhile, there are several reasons for choosing GLM in the second part. First, GLM can be directly applied on the raw cost scale, so retransformation would not be required to obtain predicted value (mean) in nominal terms, as in the case of log transformed OLS model. In addition, the mean of logged data can be very sensitive to small changes in the left tail distribution [26]. Lastly, the second regression in 2PM suffers from heteroskedasticity. In this case, GLM would be a better choice as the model allows for heteroskedasticity through the distributional family [26].

There are also common econometric issues in modeling health expenditures and health insurance. First, there is potential endogeneity with health insurance because people self-select to have insurance [33]. However, since the JKN enrolment is mandatory, this would not cause a major concern. The selection bias would be more relevant with voluntarily health insurance schemes, like private insurance [34].

Second, insurance ownership and healthcare utilization might cause perfect collinearity. Intuitively, insured people would tend to access healthcare more frequently. Nevertheless, this issue would not be entirely relevant for the Indonesian case. Demand for non-prescribed medicines and traditional medications is still quite high, although they are not covered by health insurance [35]. SUSENAS 2019 shows that this demand accounts for about 36% of annual household OOP spending in average. We did not find evidence of perfect collinearity between insurance ownership and medical use by providers.

After conducting this analysis at the national level, we then explored how much variation JKN's effect has on OOP in each region. Therefore, we further extended our analysis to the regional level by dividing the country into its six regions: Sumatera; Java & Banten; Bali & Nusa Tenggara; Kalimantan; Sulawesi; Maluku & Papua. Java and Banten is the most densely populated and most developed region, while Maluku and Papua are the least densely populated and developed regions. We replicated the same methodology and ran six regressions. However, for this regional analysis we do not disaggregate providers by public and private due to limited observations because many observations do not access inpatient or outpatient care in health facilities, especially private PHC in Eastern regions. We do not show the regression outputs. The results are available upon request.

## Results

Table 1 below shows that real annual OOP spending amounts to Rp 743,773 (US $53) on average, while 48.9% of households own only JKN as insurance and 27.2% of households have no

**Table 1. Key characteristics of study population.**

| Variable | Frequency | | Mean/Proportion |
|---|---|---|---|
| | **Unweighted** | **Weighted** | |
| **Outcome variable:** | | | |
| Real annual OOP ($) | 609.990 | 141.043.416 | $53 |
| **Explanatory variables:** | | | |
| Insurance ownership: | | | |
| Uninsured | 155.071 | 38.354.062 | 27.2% |
| JKN only | 318.768 | 69.018.564 | 48.9% |
| Private & mixed | 136.151 | 33.670.790 | 23.9% |
| Medical use*: | | | |
| Never out/inpatient | 347.491 | 78.084.982 | 55.4% |
| Outpatient only at public hospital | 8.118 | 1.701.781 | 1.2% |
| Outpatient only at private hospital | 6.186 | 1.965.959 | 1.4% |
| Outpatient only at public PHC | 65.199 | 12.520.227 | 8.9% |
| Outpatient only at private PHC | 82.361 | 22.634.513 | 16.1% |
| Outpatient only at mixed facilities | 6.768 | 1.799.909 | 1.3% |
| Inpatient only at public hospital | 19.385 | 3.686.450 | 2.6% |
| Inpatient only at private hospital | 12.360 | 3.720.778 | 2.6% |
| Inpatient only at public PHC | 6.576 | 1.357.871 | 1% |
| Inpatient only at private PHC | 3.302 | 983.939 | 0.7% |
| Inpatient only at mixed facilities | 501 | 116.758 | 0.1% |
| In & outpatient at public hospital | 7.217 | 1.500.276 | 1.1% |
| In & outpatient at private hospital | 4.798 | 1.548.654 | 1.1% |
| In & outpatient at public PHC | 4.809 | 890.682 | 0.6% |
| In & outpatient at private PHC | 2.465 | 750.537 | 0.5% |
| In & outpatient at mixed facilities | 32.454 | 7.780.100 | 5.5% |
| Health status (at least one HH member feeling sick in past one month): | | | |
| No | 238.820 | 53.330.793 | 37.8% |
| Yes | 371.170 | 87.712.623 | 62.2% |
| Location: | | | |
| Urban | 353.018 | 78.163.319 | 55.4% |
| Rural | 256.972 | 62.880.097 | 44.6% |
| Education of household head: | | | |
| At most Primary school | 317.905 | 71.481.971 | 50.7% |
| Junior high school | 93.750 | 22.380.181 | 15.9% |
| Senior high school | 121.387 | 28.130.298 | 19.9% |
| University | 76.948 | 19.050.966 | 13.5% |
| Household size | 609.990 | 141.043.416 | 3.7 |

*Definition on medical use by provider:

• SUSENAS asked whether an individual ever had a particular medical use and what is the provider.

• Households with outpatient care only means at least one of the members had outpatient care only.

• Households with inpatient care only means at least one of the members had outpatient care only.

• Households with outpatient & inpatient care means at least one of the members had outpatient and inpatient care.

insurance of any kind. Table 1 also shows that 55.4% of households did not receive any inpatient or outpatient care from 2018–19, the largest proportion for any type of medical use. There are 62.2% of households reported having a household member sick in the last month. However, among households that did not access inpatient or outpatient care, 39% has

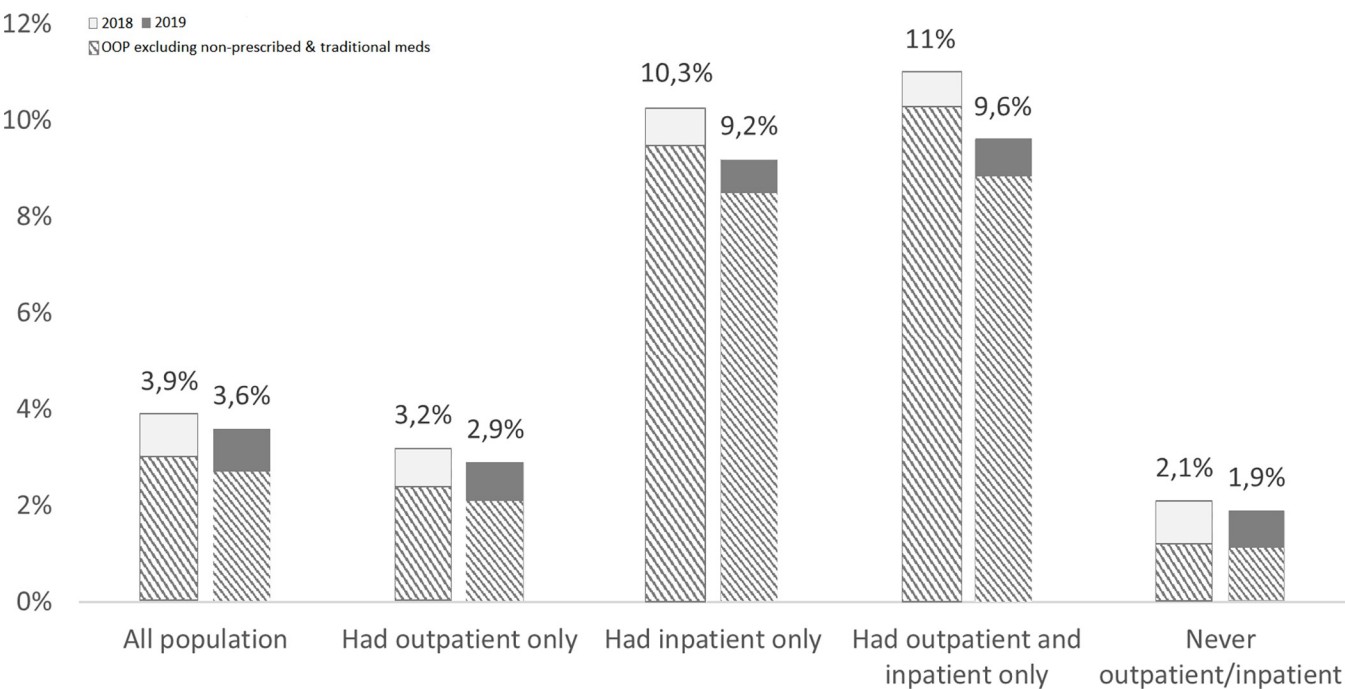

**Fig 1. Share of OOP payments out of total non-food expenditure.** Source: Own calculation using Susenas 2018 and 2019.

members that were sick in the past month. This shows that the challenge that still persists in Indonesia is that many do not access health facilities even though they are sick.

Fig 1 shows the share of OOP payments from the total of non-food expenditure. At the aggregate level, non-prescribed and traditional medicines accounted for approximately a quarter of OOP in both years. Unsurprisingly, inpatient care seems to contribute the most to the high OOP spending. Still, even households without any inpatient or outpatient care incurred some OOP (likely due to accessing medicines at shops/pharmacies). In terms of equity, our analysis found that the richest quintile pays the highest in OOP, often exceeding the threshold of catastrophic spending.

## First part–The probability of incurring OOP payment

The first part of the model shows the probability of a household incurring *any* OOP payment for healthcare. S3 Table exhibits the marginal results of this probability. JKN members have the highest probability of not having to pay any OOP (34%), while the probability for those holding private and mixed insurance is 30% and 27% for the uninsured. It is interesting to see that the uninsured have a probability of 27% to not pay OOP, which may be due to some districts' local regulation on providing healthcare support if their residents can prove their eligibility' for example Papua allows its native residents to access free healthcare at most facilities [36]. Fig 2 disaggregates this by wealth quintiles–poorer JKN members are significantly less likely to incur any OOP spending, 37% in quintile 1 to 30% in quintile 5. Getting outpatient care at a public PHC facility significantly increases the likelihood to not incur OOP at any other type of provider, especially when holding JKN (32%). There is a significantly higher probability of JKN members not paying OOP when obtaining outpatient care (33%) rather than more expensive inpatient care (21%) at PHC facilities.

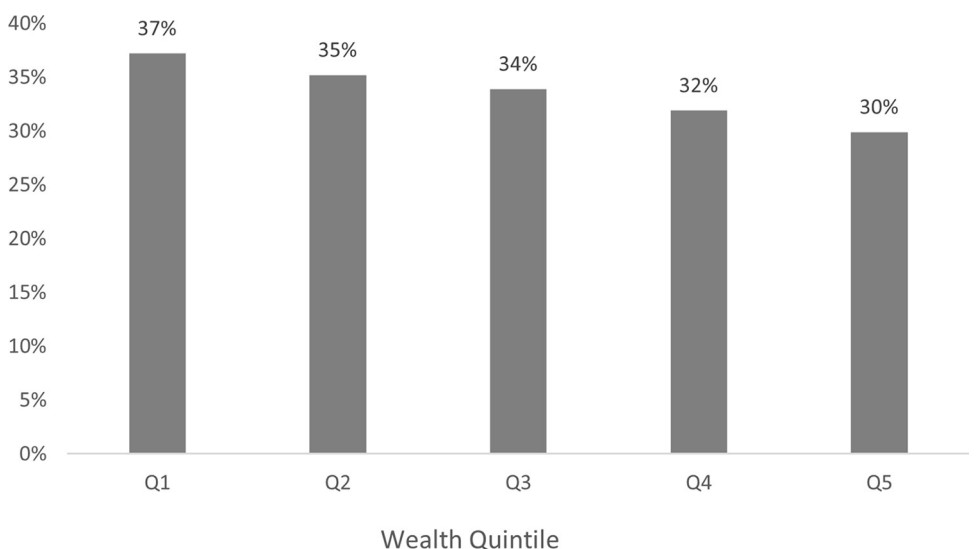

**Fig 2. JKN households' probability to obtain health care without OOP spending.** Note: *Statistically significant at 5%.

Several covariates were used as control variables, such as household size, and whether a household member was sick in the past month. If a household had at least one sick member in the past month, they have a significantly higher probability of paying OOP (6%) than households without sick members. There is also a significant increase in OOP spending as a household's size increases–the likelihood increases by 3.2% with every additional member.

## Second part–OOP payment reduction associated with JKN

The second part consists of two analyses exploring the conditional and unconditional marginal effects. The conditional effect explains *how much* OOP payment is incurred on average *if a household incurs any OOP*, thus eliminating those that did not pay OOP. The unconditional effect explains *how much OOP* is incurred on average among the *entire population*, not just those who incurred any OOP payments. S3 Table shows the marginal results where unconditional effects are obtained by multiplying the first part and the conditional effect.

**Conditional marginal effect.** The conditional marginal effect reveals that on average, households with JKN incur significantly less OOP payment compared to the uninsured (-37%) and even compared to those holding private & mixed insurance (-8%). JKN also seems to influence how much households of different wealth quintile spend on OOP–there is a significant reduction in OOP payments for the first to third quintiles compared to the uninsured (-32% to -33%). Having JKN also reduces OOP payment for all types of services across health facilities from 8% to 46%, compared to the uninsured. For example, the average OOP payment for a household receiving inpatient care at a public hospital using JKN is Rp 2,345,490 (US$ 166), which is 46% less than what an uninsured household would incur. JKN also has a strong effect on reducing OOP spending in rural areas (-35%) than in urban ones (-31%), compared to the uninsured. About 56% of Indonesians live in urban areas and the majority are in the middle-upper quintile, and they seem to have a strong preference to avoid generic medicines and prefer to pay OOP to receive quicker and better services [20,37]. These preferences often exceed JKN benefits and, thus, this segment of the population is more likely to incur OOP.

Meanwhile, the uninsured households experience 12% less OOP payments in urban areas versus rural ones. This is partly because in rural areas, delays in access to healthcare are

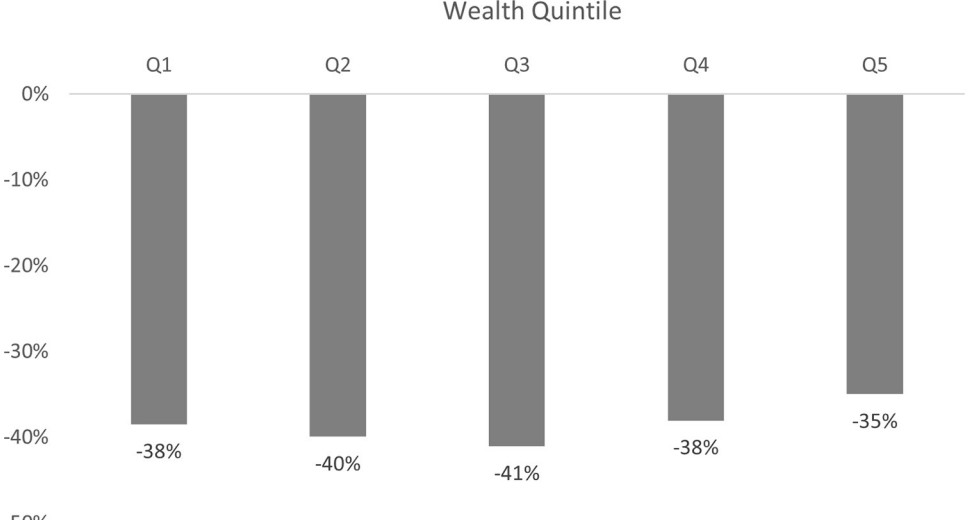

**Fig 3. JKN households' OOP payment reduction by wealth quintile, compared to uninsured HH.** Note: *Statistically significant at 5%.

persistent, causing patients to incur higher cost of treatment [38]. This delay may be caused by less availability of facilities and based on our data analysis (using Susenas 2018 and 2019), almost half of the uninsured living in rural areas are the poor (quintile 1 and 2). This is again confirmed by our analysis that the proportion of uninsured households in rural areas that did not access healthcare facilities even when they are sick are 9% higher than that who live in urban areas. On the other hand, in urban areas with higher provider density, the magnitude of supply-induced demand is weaker among poor patients [39]. For example, doctor's dual practices in urban areas will only concentrate inducement effort by self-referring to their more expensive facility, on those most likely to respond, which are those with higher ability to pay. However, in our case, the uninsured poor in urban areas account for 45,5% (quintile 1 and 2). Therefore, OOP in rural is higher than in urban among the uninsured.

**Unconditional marginal effect.** Overall, households with JKN pay 39% less on OOP compared to uninsured households, with rural areas experiencing more savings than urban areas. Fig 3 shows how much households with JKN save on OOP (on average) compared to the uninsured, broken down by wealth quintile. While those with JKN across wealth quintiles spend less on OOP for health services than the uninsured, those in Q3 save the most (41%). This means that while poorer households with JKN are more likely to obtain health care without OOP spending compared to the uninsured, the magnitude of cost savings is higher among those in the middle wealth quintile.

The next step was to observe how OOP payments changed among the different types of providers used (public/private PHC facility vs hospital) and services used (outpatient vs inpatient). At the PHC level, JKN households experience much more savings on OOP at public PHC facilities than private PHC facilities compared to uninsured households, especially for inpatient care (Fig 4).

The difference in OOP costs savings between public and private PHC facilities is much wider than the difference at the hospital level. Further analysis revealed that while JKN households have the highest probability to access health services without incurring OOP expenditure at public PHC facilities, the *magnitude* of this cost-savings is lower compared to hospitals. JKN

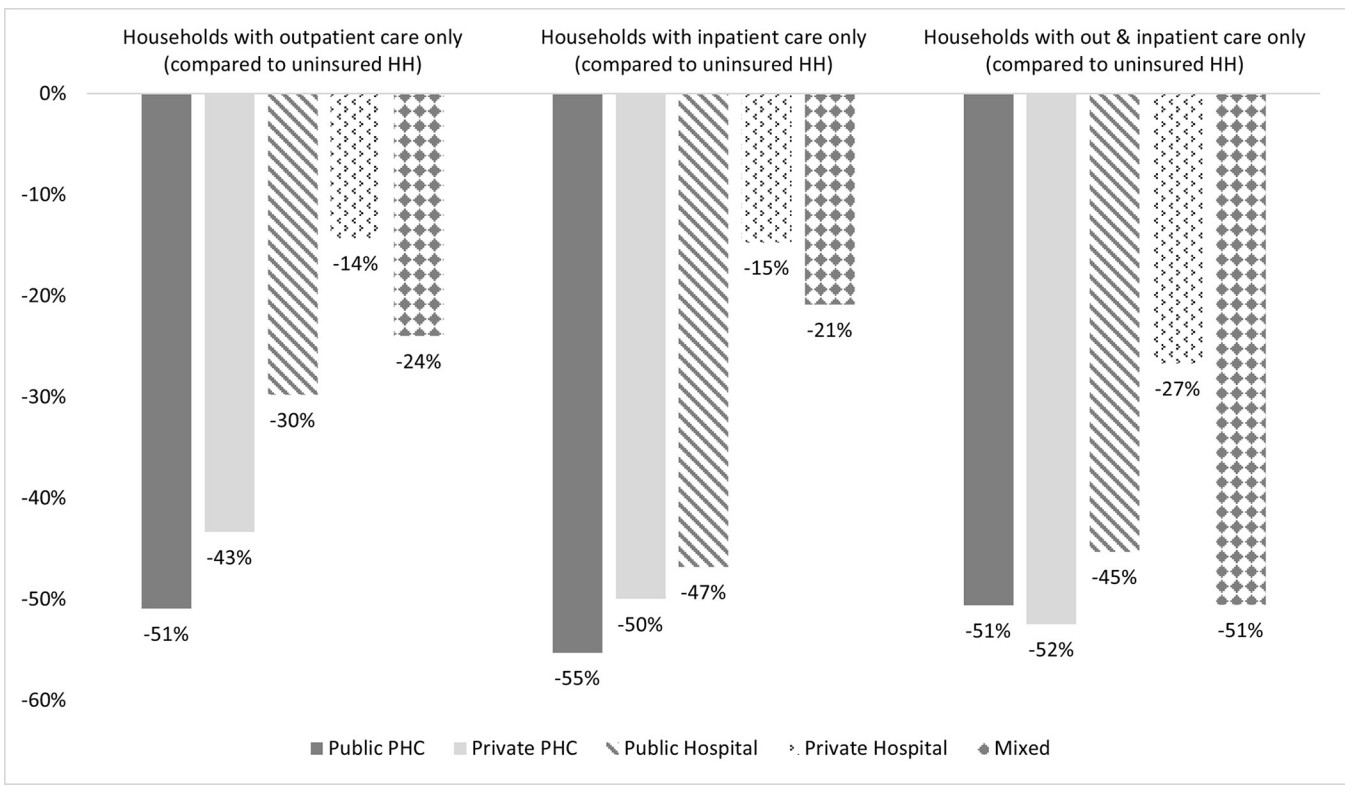

**Fig 4. JKN's effect on reducing OOP payments by facility and service type.** *Susenas asked whether an individual ever had a particular medical use and what is the health facility. It did not provide any information about frequency of medical use; Households with outpatient care only means at least one of the members had outpatient care only; Households with inpatient care only means at least one of the members had outpatient care only; Households with Outpatient & inpatient care means at least one of the members ever had outpatient and inpatient care. Households that visit multiple health facilities are not discussed; Statistically significant at 5%.

households save over 50% more in OOP at public and private hospitals than their uninsured counterparts.

We then analyzed JKN's reduction effects across Indonesia and found that cost savings are higher in the eastern, more rural part of the country. JKN coverage decreases overall OOP payment in the eastern part (Maluku and Papua, the two least developed regions) by 48% on average. Rural areas in eastern Indonesia see a reduction of 53% in OOP health spending due to JKN and urban areas see a reduction of 43%, which are much higher than the national average at 39%. Households with JKN in eastern Indonesia have the highest probability to access health services without incurring OOP spending at public PHC facilities, which households (especially the poor) use more often than in the Western, more urban parts of Indonesia [40]. Fig 5 explores how cost-savings due to JKN are distributed across the different provinces of Indonesia by type of facility and service type; provinces displayed are in sequential order from the west (Sumatera) to the East (Maluku-Papua). Broadly, we see larger OOP reductions in JKN households versus the uninsured at hospitals across regions, especially for inpatient services. The largest savings among JKN households for outpatient services at PHC facilities was in the more rural, eastern regions of Maluku and Papua (-53%), though there were significant savings for JKN households versus the uninsured across the regions for these PHC services at the more costly hospital level. As we already controlled for spatial variation of OOP health spending, differences on cost-savings magnitude are most likely related to JKN utilization.

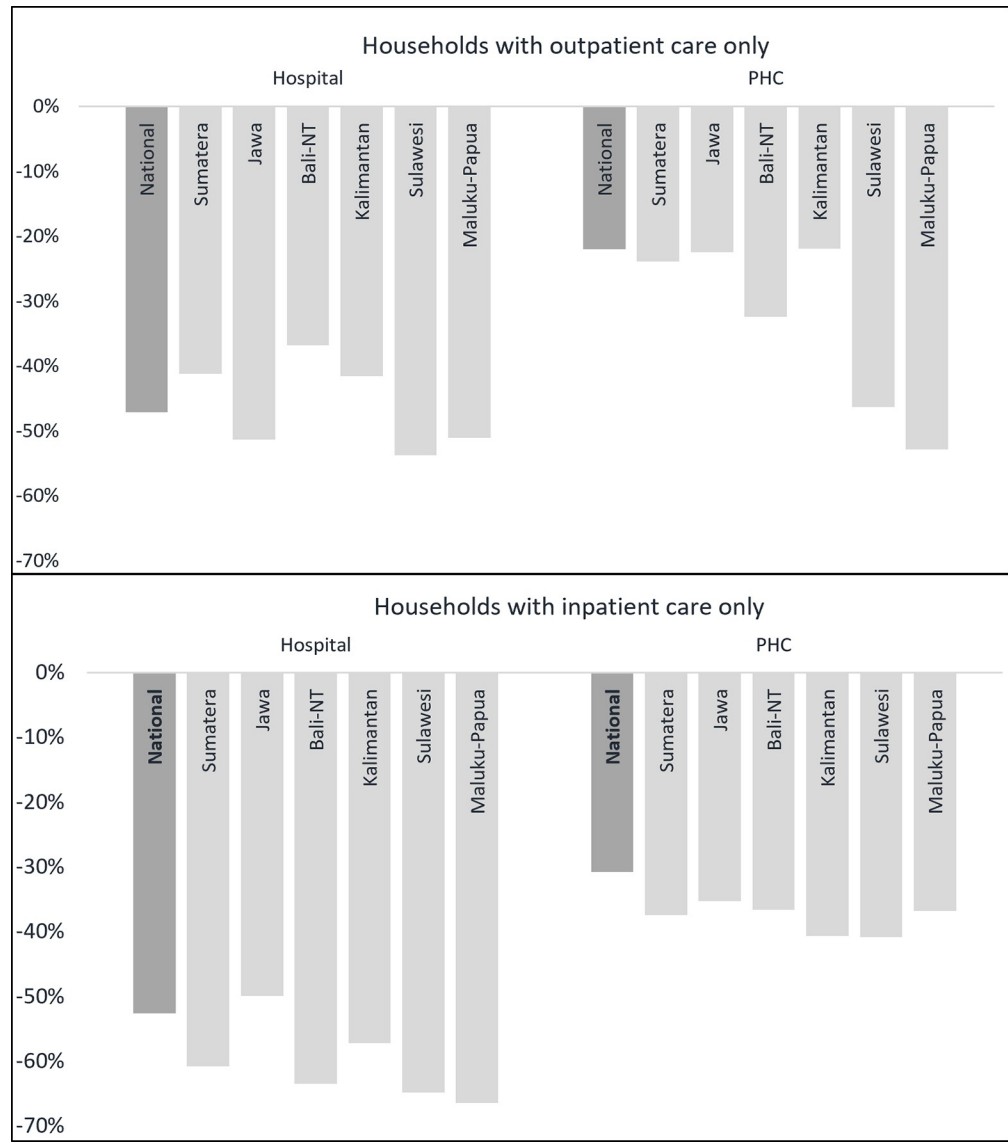

**Fig 5. JKN's effect on reducing OOP payments by facility and care type at the provincial level.** Note: *Statistically significant at 5%.

As expected, results at the national level are highly influenced by Java and Sumatera, whose observations account for 79% of the total SUSENAS sample. When broken down by wealth, JKN households in the third quintile in the capital region of Java experience the largest cost-savings by a factor of 40%. In Sumatera, where its GDP is the second highest in the nation, JKN households in the poorest quintile save 42% in OOP compared to the uninsured, while the richest quintile saves 35%. To further demonstrate JKN's impact on equity, the poorest quintile in the Maluku and Papua region sees the largest reduction (58%) in OOP spending compared to the richest quintile (28%). Similar patterns are seen in Kalimantan and Sulawesi, the regions where 53% and 61% of its residents reside in rural areas, respectively. In contrast, the regions of Java and Bali, where most higher income households reside, see less reductions in OOP health spending compared to Maluku & Papua.

## Discussion

This study demonstrates that households with JKN membership are first, far less likely than the uninsured to pay OOP when accessing health services. Second, if they do incur a cost, the magnitude of this cost is much lower among JKN households than uninsured ones. Third, this study shows that JKN has had a pro-poor benefit. JKN households in the two poorest quintiles have a higher probability to not incur *any* OOP (37% and 35% respectively) compared to those in the wealthier quintiles 4 (32%) and 5 (30%). We also found that middle-income (Q3) households saved the most in OOP costs when compared to the uninsured (41% compared to Q1's 38% and Q5's 35%). This may be due to the middle class utilizing JKN more than the poor due to better knowledge of JKN benefits, easier access to health care facilities contracted with BPJS-K., [41,42]. As economic status increases, the likelihood to obtain health services without OOP spending declines. This may be due to wealthier populations having a stronger preference for private facilities or medications that are usually not covered by JKN.

Fourth, this study found that when comparing JKN households to their uninsured counterparts, JKN members saved more at public PHC facilities than private ones. JKN households also save more (over 50%) than the uninsured at public hospitals vs private ones (at a larger magnitude at both due to steeper costs), though the difference between how much JKN members save in OOP between public and private hospitals is slimmer. There may be a larger difference of OOP savings between public and private PHC facilities because private PHC facilities are often hesitant to contract with JKN due to perceived low reimbursement rates and administrative barriers [43]. Additionally, costs for services at the PHC level are likely much more affordable for some to pay OOP versus the more expensive OOP costs that individuals can incur at the public or private hospital level [44].

Fifth and finally, this study found that JKN households in the lower economic quintiles (Q1 and Q2) living in the eastern part of Indonesia–the less urbanized and developed regions–experienced the most cost-savings. Even though a lack of service availability and understaffing remain a problem in Sulawesi and Eastern Indonesia, the population in these regions are mostly from the lower quintiles and they tend to use JKN to avoid OOP payments, when compared to the higher income groups [45]. These regions also often do not have private providers or readily accessible higher-level hospital options, so most use public PHC facilities that largely do not charge OOP. Thus, while these rural and poor populations exhibit less OOP spending, this may be more driven by supply-side factors than JKN coverage. Whereas in the more urban, affluent Western regions (like Java and Sumatera), more spend OOP in the readily available private sector and at the hospital levels [8].

These findings come with several limitations. First, our findings should be interpreted as associations rather than causality. To estimate impact, we must observe whether those owning JKN actually utilize it and benefit from it. However, this data is not available in SUSENAS' main module. To better make the case for causality, a control group could have been employed using data prior to JKN's implementation to see what changed before and after JKN, but OOP spending data in SUSENAS were only available since 2018 and that still would not serve as a perfect counterfactual. Additionally, to estimate JKN's effect among the poor and non-poor we used wealth quintiles as a proxy for PBI and Non-PBI classification. SUSENAS' main module does not estimate JKN's impact on the PBI group at the household level due to lack of knowledge whether household members own JKN. Further research should be conducted using the new health and housing module, which derives data from SUSENAS households and contains detailed information on JKN utilization. Future studies should use the data to estimate the impact of the government's subsidy towards PBI members, as this is a GoI priority. Lastly, we define a JKN household as a household in which all of its members are covered by JKN, which

might underestimate JKN's effect in reducing OOP health spending since there are households with members owning a mix of JKN and private insurance or not owning any insurance at all. Future studies could also explore how OOP payments change by service type and how supply-side readiness may influence the relationship between JKN coverage and OOP payments.

The findings from this study inform the debate on the influence of NHIs on OOP payments. In Mexico, Seguro Popular was shown to reduce catastrophic health spending by 54%, on average [6]. In China, their NHI scheme decreased OOP for inpatient care amongst the middle-aged and elderly [46]. In Thailand, during the first 10 years of its Universal Coverage Scheme, the country saw a drop in OOP payments and protected 292,000 households from impoverishment between 2004–2009 [8]. On the other hand, studies in India and Vietnam found no association between health insurance coverage and OOP payments [9,10].

Specifically in Indonesia, Aji et al conducted a study on the association of insurance on OOP prior to the JKN period in 2013, when the scheme was disaggregated by employment. The study found that Askes (insurance for civil servants) and Askeskin (insurance for the poor) households experienced 34% and 55% lower OOP, respectively [47]. Another study which utilized Susenas 2014 by Tarigan et al found that amongst the poorest that seek inpatient care, those with JKN incurred 3% less OOP than those without [48]. At the beginning of JKN implementation in 2016, an exit survey study by Pujiyanto et al found that OOP was largest amongst inpatient care patients at private hospitals [11]. A more recent study in 2020 by Nugraheni et al found JKN coverage to be associated with lower OOP for delivery and a lower risk of incurring catastrophic expenditures related to a delivery [16]. Our study confirms many of these findings and to the best of our knowledge, is the first study that analyzes how JKN influences household OOP using Susenas 2018 and 2019 and analyzes how this relationship changes by provider source, wealth quintile, and province in Indonesia. These latest Susenas surveys directly ask about a households' OOP, unlike previous versions of Susenas which may include transfers and other insurance.

Although this study shared positive findings associated with JKN enrolment, this is still in a country where the amount spent OOP (Rp 157.5 trillion in 2019) was still bigger than the amount of money spent by JKN in absolute terms (Rp 113.3 trillion in 2019) [15]. This OOP spending is largely driven by a number of factors, including the growing private sector that either doesn't contract with JKN or the willingness of wealthier populations to pay OOP, national pharmaceutical policies that still heavily promote the use of patented drugs rather than generic drugs, and the lack of emphasis on less costly public health measures and more on expensive curative services [49].

While JKN has improved the portion of public spending from 32.1% to 52.1% of total health expenditure (THE) from 2013 to 2019, this needs to continue as OOP spending still comprises 32.2% of THE in 2019, though down from 46.7% in 2012 (JKN started in 2014) [15]. This downward trend of OOP is most likely the result of JKN implementation, as private insurance schemes have been relatively constant in the last 10 years (15% of THE). In order to continue and accelerate that trend, the Indonesian government can take a series of steps. First, public PHC facilities are currently focusing on curative services, and less on promotional and preventive efforts due to the majority of JKN members being registered at these facilities. Approximately 83.2% of JKN members are registered at public PHC facilities while only 16.8% were registered at private ones in 2020 [50,51]. Therefore, workloads at public PHCs should be optimally balanced through redistributing some of their JKN members to private PHC. Consequently, there will be stronger promotive and preventive services at public PHC facilities that attract, and screen patients early could prevent higher admission to hospital, hence will have tremendous potential to reduce OOP spending [52]. Second, JKN's rates should be adjusted to further incentivize private PHC facilities to network with BPJS-K. By 2021, there were only

67% networked private PHC, in other words redistributing members may be challenging in some areas with no or little contracted private PHC. Finally, efforts to reduce OOP should be followed by contracting more private hospitals as it only accounts for 83% of total private hospital nationwide in 2020. This is important because the number of public hospitals is only 931, nearly half of the private. In order to attract more private hospital, rates should be adjusted to current prices as they have not been updated in the past 6 years.

This study revealed that JKN has a positive effect on shielding its members, especially the poorest ones, from expensive OOP costs. This finding shows that JKN is making strides to meet its original equity objectives when it was established in 2014 and adds to the growing body of literature globally about the protective effects of an NHI scheme with wide coverage on costly OOP payments experienced by the population. The study suggests that better engagement of the private sector and increased investment in supply-side infrastructure, especially in the Eastern provinces, can further help the GoI secure financial protection of its population on accessing health services. Thus, while there is still progress to be made in Indonesia to reduce OOP spending and improve equity, their NHI scheme seems to be moving the country in the right direction on its path to UHC.

## Supporting information

**S1 Table. Descriptive statistics by insurance types.** This table is the extended version of Table 1, which observes 3 household population segments; uninsured, JKN, and private and mixed insurance (households whose members has a variety of insurance, e.g: JKN and private insurance) with both weighted and unweighted totals. These segments are broken down by all variables used in the regression model.
(DOCX)

**S2 Table. Two part model output.** This table depicts the regression output of the two-part model, where the first part is a logit model with the binary dependent variable of zero and positive OOP payments. It estimates the likelihood of a household incurring zero or positive OOP payments. The second part is a Generalized Linear Model (GLM) with gamma error distribution and a log link function. It estimates the level or intensity of OOP payment, conditional on a household spending anything OOP.
(DOCX)

**S3 Table. Marginal results, key variables.** This table shows the marginal results of the regression output; the first part, and the second part which consists of conditional and unconditional effects. The marginal result of the first part shows households' probability of paying OOP, the conditional effect shows the average OOP spending conditional on households having spent on OOP, and the unconditional effect shows the estimated effect on the entire population.
(DOCX)

## Acknowledgments

The authors would like to express to Taufik Hidayat and Ardi Adji from the National Team for the Acceleration of Poverty Reduction (TNP2K) for their inputs in the analysis process. We also thank Professor Hasbullah Thabrany, Dr. Trihono, Halimah Mardani, Dr. Nirmala Ravishankar, and Jack Langenbrunner for assisting with this manuscript.

## Author Contributions

**Conceptualization:** Nirwan Maulana, Prastuti Soewondo, Nadhila Adani, Paulina Limasalle, Anooj Pattnaik.

**Data curation:** Nirwan Maulana, Nadhila Adani.

**Formal analysis:** Nirwan Maulana, Nadhila Adani, Anooj Pattnaik.

**Funding acquisition:** Anooj Pattnaik.

**Investigation:** Nirwan Maulana, Nadhila Adani, Anooj Pattnaik.

**Methodology:** Nirwan Maulana, Anooj Pattnaik.

**Project administration:** Prastuti Soewondo, Nadhila Adani, Paulina Limasalle, Anooj Pattnaik.

**Supervision:** Prastuti Soewondo, Anooj Pattnaik.

**Validation:** Prastuti Soewondo.

**Visualization:** Nirwan Maulana, Nadhila Adani, Anooj Pattnaik.

**Writing – original draft:** Nirwan Maulana, Prastuti Soewondo, Nadhila Adani, Anooj Pattnaik.

**Writing – review & editing:** Nirwan Maulana, Nadhila Adani, Paulina Limasalle, Anooj Pattnaik.

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
