## [Decision Letter · Decision Letter 0]

7 Sep 2021

PGPH-D-21-00150

How Jaminan Kesehatan Nasional (JKN) coverage influences out-of-pocket (OOP) payments by vulnerable populations in Indonesia

Dear Dr. Maulana,

Thank you for submitting your manuscript to PLOS Global Public Health. After careful consideration, we feel that it has merit but does not fully meet PLOS Global Public Health’s publication criteria as it currently stands. Therefore, we invite you to submit a revised version of the manuscript that addresses the points raised during the review process.

We look forward to receiving your revised manuscript.

Kind regards,

Habib Hasan Farooqui, MD

Academic Editor

Journal Requirements:

Additional Editor Comments (if provided):

Understanding the impact of publicly funded insurance scheme is an important area of policy research and provides insights into the performance of the scheme which may help implementing agencies and policy makers to address implementation and policy bottlenecks. The authors have made a good attempt to address the question - How Jaminan Kesehatan Nasional (JKN) coverage influences out-of-pocket (OOP) payments by vulnerable populations in Indonesia? The methodology section and statistical analysis requires major revision. In addition, to understand the insurance scheme design, eligibility criteria, health system context and implementation and impact pathways detailed account of scheme should be provided. Also, the outcome indicator out of pocket payments (OOP) needs to be clearly defined and reported as per the published international literature.

Reviewers' comments:

Reviewer's Responses to Questions

**Comments to the Author**

1. Does this manuscript meet PLOS Global Public Health’s publication criteria? Is the manuscript technically sound, and do the data support the conclusions? The manuscript must describe methodologically and ethically rigorous research with conclusions that are appropriately drawn based on the data presented.

Reviewer #1: Partly

Reviewer #2: Partly

2. Has the statistical analysis been performed appropriately and rigorously?

Reviewer #1: Yes

Reviewer #2: Yes

3. Have the authors made all data underlying the findings in their manuscript fully available (please refer to the Data Availability Statement at the start of the manuscript PDF file)?

Reviewer #1: Yes

Reviewer #2: Yes

4. Is the manuscript presented in an intelligible fashion and written in standard English?

Reviewer #1: Yes

Reviewer #2: No

5. Review Comments to the Author

Reviewer #1: The paper needs some significant modification, for it to be published. There are problems in choosing the key outcome variable(s) and suggested change might potentially change the entire results. However, the study has its significant and needs to be published with modifications.

Specific comments:

Please mention the unweighted sample size (pp4, line112)

Exclusion of non-prescription medicines and traditional medicine from overall assessment of OOP provides an incomplete picture of OOP. Literature suggest that it is the poorer sections of the society and those living in rural areas would be using these types of care more than the richer section living in urban areas, working in the formal sector. Thus in figure 1 where share of OOP as % of household non-food expenditure is being captured, these expenses need to be included to provide a comprehensive expenditure and results are biased against the poorer sections, showing lower hardship on their part. The author could provide a comparative picture of OOP including and excluding these elements, while limiting to OOP incurred in modern facilities for regression results.

It is important to know the extent of free care accessed by households, as this is a key goal of the JKN program.

However, treating all other expenses as same (considering them as positive as part of the binary outcome variable) is a major limitation of the study, as this fails to capture the effect of the scheme in reducing financial hardship. Similarly, absolute OOP also does not make much sense unless adjusted for households' ability to pay. Financial hardship can be better assessed if the authors had taken catastrophic incidences (at various thresholds) as these are adjusted for households' ability to pay.

A stated limitation of the SUSENAS data is it doesn't provide any information on the intensity of use of hospitalization, times used, days of stay etc. It needs to be noted that there exists possibilities that the same family might use different types of facilities for different incidence of hospitalization and hence have different outcomes of OOP. this would significantly affect the results. This needs to be specified.

The characteristics of households who are covered under the JKN and who are not could be very different, thus creating selection biases. It is important that the authors provide a summary table, stating the socio-economic characteristics of households based on coverage under various insurance schemes and compare those who are not covered.

Reviewer #2: Manuscript ID: PGPH-D-21-00150

Manuscript Title: How Jaminan Kesehatan Nasional (JKN) coverage influences out-of-pocket (OOP) payments by vulnerable populations in Indonesia

The above manuscript makes new attempt at capturing the impact of JKN (national health insurance) on households’ OOP spending in Indonesia. This was made possible by an improved household tool capturing OOP payment in SUSENAS (National Socio-Economic Survey) beginning from 2018.

Following are the observations that the authors may want to address to improve the quality of the manuscript.

Scheme design: JKN scheme design is unclear from the reading of the manuscript. For instance, are there exclusion or inclusion criteria in the scheme (number of household members included or excluded from the scheme) and the scheme benefit is unknown:) i) What benefits are provided (benefit package that contains only inpatient and outpatient or only inpatient)? Ii) how many clinical/medical interventions are included in the scheme?; iii) are there caps imposed per households per annum on the benefits? Iv) Are medications provided free? V) And what are additional benefits added during the implementation of JKN as against pre-JKN era, especially in the public facilities? vi) If benefits are comprehensive, do these facilities use negative list (benefits not covered) or positive list (number of packages covered by the scheme).

Methods: it outlines sample size indicating 263,666,217 individuals (69,954,912 households). Although it correctly identifies this number to be sample weighted numbers, the key sample size that must be reported is about sample size without weights. Otherwise, it gives an impression that as if the entire Indonesian population is sampled in the survey, which is nothing short of a census, and that is not the case here. And are these samples collected in one period (March) is the total sample or half of the annual sample?

The methods section did not clearly spell out whether the survey captured information about inpatient and outpatient expenditure separately. If so, what was the recall period for Inpatient and Outpatient? Did the survey also collect information about non-medical expenditure including lodging, transport, etc. because these expenditure can be considerably higher among those in inaccessible and underprivileged areas, an objective which JKN seeks to bridge inequity.

Statistical Analysis: Page 9: If most part of the population are covered by the sample, and assuming such a large sample, it is unlikely that the authors did not obtain adequate samples to disaggregate providers by public and private – due to limited observations. This needs explanation. It is equally important to consider the fact that even after being covered by health insurance, households do end up paying out of pocket, especially in private facilities.

Results:

1. Page 11 - First Part Model Results: What accounts or reasons for an uninsured (27%) not paying OOP despite being ill. It is unclear whether all of those uninsured utilised public facilities and for free? This requires explanation.

2. Page 12: Despite being covered, what explains JKN enrolled population still paying OOP, even though they may be paying less than an uninsured?

3. Page 12: Despite being uncovered, what explains uninsured to spend less in urban than in rural areas?

4. Page 13: What factors/reasons contribute to middle wealth quintile groups to obtain higher cost savings compared to poor and near-poor JKN population?

5. Page 13: If the benefit package and the cost associated with it are borne by JKN, what explains JKN households to experience more savings on OOP at public PHC facilities than private PHC facilities, especially for inpatient care? Can PHC handle the bulk of inpatient care? Table 1 indicated that only 1% of population accessed PHC inpatient services? If that is so, how important is this result?

6. In general, overall reduction in OOP in Indonesia from 46% in 2012 to 32% (as % to THE) in 2019 may be the result of JKN and a large expansion of private health insurance as well. This needs to be observed and reported.

Language:

Page 7: Statistical Analysis: First line: This line appears to provide a wrong message. Rather “JKN ownership on OOP health spending” should have been “JKN enrolment on OOP health spending”. The literature in this area specifies insurance enrolees rather than as owners of insurance.

Others

Page 9: Is East Timor part of Indonesia or West Timor? Please check the second paragraph.

6. PLOS authors have the option to publish the peer review history of their article (what does this mean?). If published, this will include your full peer review and any attached files.

**Do you want your identity to be public for this peer review?** For information about this choice, including consent withdrawal, please see our Privacy Policy.

Reviewer #1: **Yes: **Indranil Mukhopadhyay

Reviewer #2: **Yes: **Sakthivel Selvaraj

---

## [Decision Letter · Decision Letter 1]

7 Apr 2022

PGPH-D-21-00150R1

How Jaminan Kesehatan Nasional (JKN) coverage influences out-of-pocket (OOP) payments by vulnerable populations in Indonesia

Dear Dr. Maulana,

Thank you for submitting your manuscript to PLOS Global Public Health. After careful consideration, we feel that it has merit but does not fully meet PLOS Global Public Health’s publication criteria as it currently stands. Therefore, we invite you to submit a revised version of the manuscript that addresses the points raised during the review process.

We look forward to receiving your revised manuscript.

Kind regards,

Habib Hasan Farooqui, MD

Academic Editor

Journal Requirements:

Additional Editor Comments (if provided):

I would appreciate if the authors could provide regression tables and modify the paper addressing the reviewers comments.

Please find below reviewers comments for your consideration.

Reviewer 1: The authors have not responded to all previous comments satisfactorily. For instance, using 'OOP expenses' without trying to adjust it by the ability to pay of the household (income/household consumption expenditure) remains a key limitation of the study. Despite pointing it out, the authors have not addressed this.

Though the authors have explained the two part model they have used in the methods section, they have not provided the regression results. In the absence of regression results it is not possible to verify the statements or observations made. the current paper essentially demonstrates the cross tabs and related figures, which are not adequate to support the results and subsequent discussions, particularly related to establishing any association!

The authors have not explained the logic of using two rounds of SUSENAS survey and the methods applied to use the two rounds. Remains a major limitation of the study.

There are contradictions in the results presented and policy recommendations made. For instance in line 476 the authors argue that Private hospitals should focus on curative services, while reporting repeatedly that OOP is much higher in private facilities!

Some specific issues:

Line 174: the variable "at least one household member feeling sick in the past month" "can be a proxy controlling for chronic illness"! This is surprising! needs better explanation!

line 298: The unconditional effect should take into account households who have any event of illness, rather than all households

line 310: "middle-upper quintile, and they seem to have a strong preference to avoid generic medicines..." such statements, if not drawn from the study findings, need to be backed up by references.

LIne 313: Urban areas have lower OOP because of more facilities...: such observations are contradictory to the vast literature on induced demand which shows that more facilities and greater ability to pay increases prices in urban areas. need references.

Reviewer 2: The authors must be complimented for diligently improving an earlier version and responding to most comments. The manuscript has really shaped up well.

Reviewers' comments:

Reviewer's Responses to Questions

**Comments to the Author**

1. If the authors have adequately addressed your comments raised in a previous round of review and you feel that this manuscript is now acceptable for publication, you may indicate that here to bypass the “Comments to the Author” section, enter your conflict of interest statement in the “Confidential to Editor” section, and submit your "Accept" recommendation.

Reviewer #1: (No Response)

Reviewer #2: All comments have been addressed

2. Does this manuscript meet PLOS Global Public Health’s publication criteria? Is the manuscript technically sound, and do the data support the conclusions? The manuscript must describe methodologically and ethically rigorous research with conclusions that are appropriately drawn based on the data presented.

Reviewer #1: Partly

Reviewer #2: Yes

3. Has the statistical analysis been performed appropriately and rigorously?

Reviewer #1: I don't know

Reviewer #2: Yes

4. Have the authors made all data underlying the findings in their manuscript fully available (please refer to the Data Availability Statement at the start of the manuscript PDF file)?

Reviewer #1: No

Reviewer #2: Yes

5. Is the manuscript presented in an intelligible fashion and written in standard English?

Reviewer #1: Yes

Reviewer #2: Yes

6. Review Comments to the Author

Reviewer #1: The authors have not responded to all previous comments satisfactorily. For instance, using 'OOP expenses' without trying to adjust it by the ability to pay of the household (income/household consumption expenditure) remains a key limitation of the study. Despite pointing it out, the authors have not addressed this.

Though the authors have explained the two part model they have used in the methods section, they have not provided the regression results. In the absence of regression results it is not possible to verify the statements or observations made. the current paper essentially demonstrates the cross tabs and related figures, which are not adequate to support the results and subsequent discussions, particularly related to establishing any association!

The authors have not explained the logic of using two rounds of SUSENAS survey and the methods applied to use the two rounds. Remains a major limitation of the study.

There are contradictions in the results presented and policy recommendations made. For instance in line 476 the authors argue that Private hospitals should focus on curative services, while reporting repeatedly that OOP is much higher in private facilities!

Some specific issues:

Line 174: the variable "at least one household member feeling sick in the past month" "can be a proxy controlling for chronic illness"! This is surprising! needs better explanation!

line 298: The unconditional effect should take into account households who have any event of illness, rather than all households

line 310: "middle-upper quintile, and they seem to have a strong preference to avoid generic medicines..." such statements, if not drawn from the study findings, need to be backed up by references.

LIne 313: Urban areas have lower OOP because of more facilities...: such observations are contradictory to the vast literature on induced demand which shows that more facilities and greater ability to pay increases prices in urban areas. need references.

Reviewer #2: The authors must be complimented for diligently improving an earlier version and responding to most comments. The manuscript has really shaped up well.

7. PLOS authors have the option to publish the peer review history of their article (what does this mean?). If published, this will include your full peer review and any attached files.

**Do you want your identity to be public for this peer review?** For information about this choice, including consent withdrawal, please see our Privacy Policy.

Reviewer #1: **Yes: **Indranil Mukhopadhyay

Reviewer #2: **Yes: **Sakthivel Selvaraj

---

## [Editor Report · Decision Letter 2]

9 Jun 2022

How Jaminan Kesehatan Nasional (JKN) coverage influences out-of-pocket (OOP) payments by vulnerable populations in Indonesia

PGPH-D-21-00150R2

Dear Mr maulana,

We are pleased to inform you that your manuscript 'How Jaminan Kesehatan Nasional (JKN) coverage influences out-of-pocket (OOP) payments by vulnerable populations in Indonesia' has been provisionally accepted for publication in PLOS Global Public Health.

Best regards,

Habib Hasan Farooqui, MD

Academic Editor